# Identifying novel prodromal symptoms of eclampsia: A two-country, case-control study

Roxanne Hastie[1,2], Farhatulain Ahmed[3], Parinaz Mehdipour[1,2,4], Bernard Yan[5,6], Susan P. Walker[1,2], Jacqui Visser[7], Anam Bashir[3], Lyle Gurrin[8], Anthea Lindquist[1,2], Jessica A. Atkinson[1,2], Catherine Cluver[1,7☯], Lina Bergman[7,9☯], Stephen Tong[1,2☯]*

1 Mercy Perinatal, Mercy Hospital for Women, Heidelberg, Victoria, Australia, 2 Department of Obstetrics, Gynaecology and Newborn Health, University of Melbourne, Melbourne, Victoria, Australia, 3 Maternal Fetal Medicine Unit, Fatima Memorial Hospital, Lahore, Punjab, Pakistan, 4 Biostatistics Unit, Faculty of Health, Deakin University, Burwood, Victoria, Australia, 5 Department of Neurology, Royal Melbourne Hospital, Melbourne, Australia, 6 Department of Medicine, University of Melbourne, Melbourne, Australia, 7 Department of Obstetrics and Gynaecology, Stellenbosch University, Cape Town, South Africa, 8 Melbourne School of Population and Global Health, University of Melbourne, Melbourne, Australia, 9 Department of Obstetrics and Gynaecology, University of Gothenburg, Gothenburg, Sweden

☯ These authors contributed equally to this work.
* stong@unimelb.edu.au

## Abstract

### Background

Magnesium sulphate halves the risk of eclampsia. There is no consensus on who to give magnesium sulphate prophylaxis because clinical tools are poor at identifying those at risk. Known prodromal symptoms such as headache, visual disturbance, or epigastric pain have modest associations with eclampsia. We set out to identify new prodromal symptoms of eclampsia.

### Methods and findings

This case-control study prospectively recruited participants in South Africa and Pakistan who had eclampsia, preeclampsia, or normotensive pregnancies. We asked whether they experienced 20 neurological symptoms, within 7 days of the seizure for those who had eclampsia. The primary analysis was the likelihood of symptoms occurring before eclampsia, compared to being present with preeclampsia. 341 participants were recruited with eclampsia, 1,355 with preeclampsia and 389 with normotensive pregnancies. When comparing symptoms among those who had eclampsia versus preeclampsia, the odds ratios (OR) were 2.56 (95% confidence interval (CI) [1.81,3.62]; $p < 0.001$) for headache, 5.73 (95% CI [4.44,7.39]; $p < 0.001$) for visual disturbances and 2.25 (95% CI [1.76,2.89]; $p < 0.001$) for epigastric pain. We identified 10 symptoms with odds ratios over 10 for eclampsia. Odds ratios were 42.03 (95% CI [23.66,74.68]; $p < 0.001$) for twitching/jerking limbs (30.5% eclampsia versus 1% preeclampsia); 36.00 (95% CI [18.34,70.65]; $p < 0.001$) for affected hearing (21.1% versus 0.7%)' 33.60 (95%

**Data availability statement:** All relevant data are within the manuscript and its Supporting information files.

**Funding:** This study was supported by the RANZCOG Women's Health Foundation https://ranzcog.edu.au/womens-health/foundation/. R.H. (#1176922), S.T. (#2017897) and A.L. (#2036158) were all supported by the National Health and Medical Research Council (https://www.nhmrc.gov.au). L.B. is a Wallenberg Scholar, supported by the Wallenberg Center for Molecular and Translational Medicine (https://www.gu.se/en/molecular-translational-medicine). C.C. receives salary support from the Mercy Health Foundation (https://www.mercyhealthfoundation.org.au). The funders had no role in study design, data collection and analysis, decision to publish, or preparation of the manuscript.

**Competing interests:** I have read the Journal's policy and authors of this manuscript have the following competing interests: ST, CC, LB, and SW provide consultancy services for Diamedica Therapeutics. CC and ST provide consultancy services for Avilar Therapeutics, Beech Biotech, and Gondola Bio. Merck Darmstadt provide drug and placebo for 2 current treatment trials for preeclampsia where LB and CC are principal investigators. RH is a statistical reviewer for PLOS Medicine. LB has received reagents for PlGF and sFlt-1 from Thermo Fisher, Revvity and Roche for work in prediction of preeclampsia and adverse outcomes.

**Abbreviations:** ACOG, American College of Obstetricians and Gynecologists; CI, confidence interval; OR, odds ratios; REDCap, Research Electronic Data Capture.

CI [21.39,52.78]; $p<0.001$) for affected mind state (38.7% versus 1.8%); 33.12 (95% CI [19.46, 54.37]; $p<0.001$) for impaired speech; 23.71 (95% CI [16.49,34.10]; $p<0.001$) for feelings of doom; 26.59 (95% CI [7.82,90.41]; $p<0.001$) for severe vertigo; 20.52 (95% CI [14.22,29.63]; $p<0.001$) for confusion; 18.16 (95% CI [10.76,30.66]; $p<0.001$) for jitters; 15.18 (95% CI [11.34,20.33]; $p<0.001$) for difficulty concentrating; and 10.49 (95% CI [6.76,16.27]; $p<0.001$) for weakness/paralysis. These symptoms were rare among normotensive pregnancies. Only 2.4% of women with eclampsia did not experience any prodromal symptoms. This study is limited by the fact that we asked about prodromal symptoms after the seizure happened, and the potential for recall bias.

## Conclusions

Ten prodromal symptoms exhibit far stronger associations with eclampsia than headache, visual disturbances, or epigastric pain. Eclampsia is uncommon without any prodromal symptoms. It may be useful to screen these symptoms among women with preeclampsia as part of clinical history taking to guide management. They could help direct magnesium sulphate prophylaxis to those with a higher risk of eclampsia.

### Author summary

#### Why was this study done?

- Eclampsia is a severe pregnancy complication where women with preeclampsia suffer life-threatening seizures.

- Giving magnesium sulphate can half the risk of eclampsia, but predicting who is at risk of an eclamptic seizure is challenging.

- Headache, visual changes, and upper abdomen pain are widely used by clinicians as early warning signs that a seizure might happen, but their predictive accuracy is modest.

- We set out to find novel symptoms with stronger links to eclampsia than those currently known.

#### What did the researchers do and find?

- We performed a study in South Africa and Pakistan, in which we asked women who had a recent eclamptic seizure which symptoms they experienced (before the seizure occurred), among a list of novel neurological symptoms, which have not been reported to be associated with eclampsia

- Our research discovered 10 new symptoms with far stronger associations with eclampsia than currently known symptoms.

- The strength of the association between the 10 new symptoms and eclampsia was similar between South Africa and Pakistan.

PLOS Medicine

**What do these findings mean?**

- We have identified new symptoms that may predict eclampsia better than the current symptoms widely used in the clinic.

- They may be useful as part of history taking to identify who is at increased risk of eclampsia and could benefit from magnesium sulphate to prevent it from happening.

- As we asked about symptoms after the seizure occurred rather than before, it may be worthwhile for our findings to be validated.

## Introduction

Eclampsia is a tonic-clonic convulsion arising from neurological injury caused by severe preeclampsia [1]. With a prevalence of 1–2% among women with preeclampsia [2], it can cause maternal death [3], serious morbidity such as intracerebral haemorrhage [4], or persisting neurological deficits [5]. While more frequent in low- and middle-income countries (50–151/10,000 births), it also occurs in higher resource settings (1–10/10,000 births) [1,6].

Headache, visual changes, and epigastric pain are prodromal symptoms ubiquitously used as part of clinical history among women with preeclampsia to infer eclampsia risk. The presence of these symptoms means preeclampsia should be considered 'preeclampsia with severe features', according to the American College of Obstetricians and Gynaecologists Guidelines [7]. Furthermore, those who develop these symptoms may be offered magnesium sulphate because this medication reduces the risk of eclampsia by 58% [2] and may reduce maternal death [2,8]. However, a meta-analysis concluded these symptoms have modest likelihood ratios in predicting eclampsia [9]. The likelihood ratios for eclampsia risk were 2.25 (95% confidence interval (CI) [0.61,1.03]) for headache; 5.81 (95% CI [1.74,19.42]) for visual disturbance and 3.4 (95% CI [1.02,11.31]) for epigastric pain. Furthermore, small studies have reported that 20% [10] or 41% [11] of eclamptic seizures occur without these prodromal symptoms.

We set out to find new prodromal symptoms of eclampsia that may have stronger associations than these existing symptoms that are widely used in the clinic. To do this, we undertook a large prospective study running simultaneously in Pakistan and South Africa to characterise the association between 20 neurological symptoms and their occurrence before the onset of eclampsia. We considered the known prodromal symptoms of headache, visual changes, and epigastric pain. But we also screened for symptoms not previously linked with this condition, but occur with other neurological conditions where symptoms might originate from central nervous system pathology, such as posterior reversible encephalopathy syndrome [12]. We compared their likelihood of being present before an eclamptic seizure versus being present among women with preeclampsia who did not experience eclampsia. We also determined how often they occurred in normotensive pregnancies.

## Methods

### Study design, setting, and population

This multi-site, case-control study prospectively recruited women across three tertiary referral centres: Fatima Memorial Hospital and Jinnah Hospital in Lahore, Pakistan; and Tygerberg Hospital in Cape Town, South Africa. Tygerberg Hospital is a state funded referral hospital in the Western Cape Provence of South Africa. The hospital serves a population of 3.4 million people and delivers more than 8,000 high-risk pregnancies annually. Both Fatima Memorial and Jinnah Hospitals are tertiary level hospitals based in Lahore, Pakistan, that care for low- and high-risk pregnancies. Both hospitals provide care for 6,000–8,000 births annually. Between 2018 and 2023, we recruited pregnant or postpartum women who experienced eclampsia within the last 7 days, women with preeclampsia who had not experienced eclampsia in the current pregnancy and women with normotensive pregnancies. Eclampsia was defined as the new onset of tonic-clonic seizures

among pregnant or newly postpartum women with preeclampsia. Preeclampsia was defined according to the International Society for the Study of Hypertension in Pregnancy [13] as hypertension (>140 mmHg systolic blood pressure or >90 mmHg diastolic blood pressure) after the 20th week of gestation and the presence of one or more of the following new onset conditions: proteinuria, renal insufficiency (creatinine >90 μmol/L), liver involvement (elevated transaminases and/or severe right upper quadrant or epigastric pain), neurological complications, haematological complications (thrombocytopenia, disseminated intravascular coagulation, haemolysis), or foetal growth restriction. In the South African cohort, proteinuria had to be present for a diagnosis of preeclampsia. Normotensive pregnant or postpartum women were those without hypertension, gestational hypertension, or eclampsia. We did not include women with multifetal pregnancies, those with seizures attributed to a diagnosis other than eclampsia (such as central nervous system infections, a history of seizures or epilepsy) and women unable to provide informed consent. To avoid crossover, we excluded women with preeclampsia who were enrolled and undertook the survey but subsequently developed eclampsia, and we excluded those who were enrolled as normotensive when they undertook the survey but who later developed a hypertensive disorder or eclampsia.

All the authors contributed to the study design, vouch for the completeness and accuracy of the data, and have agreed to submit the manuscript for publication. The senior and first authors wrote all drafts of the manuscript. The protocol was shared with an Egyptian team who are undertaking an independent investigation of these symptoms.

All relevant data are available within the manuscript and supporting information file (S2 File)

## Procedures

Women presenting to each site with eclampsia were identified by the research team. Women with preeclampsia and normotensive pregnant women were recruited among those seeking maternity services. When clinically stable, women were approached about the study and invited to participate. Those willing and able to provide informed consent were enrolled into the study.

At enrolment, all participants were asked a series of standardised structured interview questions (translated to the local language at each site) about symptoms they experienced prior to seizure onset for those with eclampsia and prior to enrolment for women with preeclampsia and normotensive pregnancies. The symptoms investigated were those previously associated with eclampsia: headache, epigastric pain, visual disturbances, and nausea and vomiting. Other symptoms investigated were those typically associated with neurological disorders outside of pregnancy including memory loss, difficulty concentrating, confusion, mood changes, twitching or jerking of limbs, hearing or speech impairment, muscle weakness or paralysis. The complete list of symptoms screened in our survey is listed in Table A in S1 File.

Maternal demographics and clinical characteristics were also obtained from maternal interview and review of medical records. Participants were followed from enrolment until discharge from hospital with birthing and maternal and neonatal outcome data obtained. All data was captured using the secure online database Research Electronic Data Capture (REDCap).

## Statistical analysis

Statistics summarising the distribution of clinical characteristics were prepared for women with eclampsia, preeclampsia and those with normotensive pregnancies.

As per our a priori protocol (S1 Protocol), the primary analysis compared the prevalence of prodromal symptoms between women with eclampsia to those with preeclampsia. The magnitude of the association between each prodromal symptom and the likelihood of eclampsia was estimated as an odds ratio (with 95% confidence intervals) from a univariable logistic regression model.

A similar analysis was performed comparing the prevalence of prodromal symptoms between women with eclampsia and normotensive women. The frequency and number of symptoms were also compared qualitatively between women

with eclampsia and i) preeclampsia and ii) normotensive women without formal statistical testing. Analyses were performed using Stata (version 18.0) and R (version 4.4.3).

In a post-hoc analysis, a complete case multivariable logistic regression was used to estimate adjusted odds ratios, controlling for maternal age, body mass index, and country of recruitment.

### Ethical approval

Local ethical approval was obtained from Fatima Memorial Hospital College of Medicine and Dentistry, Lahore, Pakistan, Institutional Review Board (FMH-05-2018-IRB-448-M) and Stellenbosch University, Cape Town: protocol number N18/03/034; Federal Wide Assurance number 00001372; Institutional Review Board approval number IRB0005239. All participants or their guardians provided signed informed consent.

This study is reported as per the Strengthening the Reporting of Observational Studies in Epidemiology (STROBE) guideline (S1 STROBE Checklist).

### Results

Between 2018 and 2023, we recruited 2,142 participants. 341 had eclampsia, 1,355 had preeclampsia and 389 had normotensive pregnancies. Among those with preeclampsia, we excluded 23 who developed eclampsia after recruitment as they represent a cross over cohort. Similarly, among normotensive women 22 developed hypertension or preeclampsia and were also excluded. Additionally, 12 women with eclampsia were recruited more than 7 days after experiencing an eclamptic seizure and were excluded from analysis (Fig 1). Of those recruited during pregnancy, gestational ages at recruitment were similar among all cohorts. Compared to those who had preeclampsia or normotensive pregnancies those who had eclampsia were younger, had a lower body mass index, were more often nulliparous or booked later for antenatal care, or did not receive any antenatal care (Tables 1 and B in S1 File).

1,453 (69.7%) were recruited from Cape Town, South Africa, and 632 (30.3%) from Lahore, Pakistan (baseline information by country shown on Table C in S1 File). Participants from the two countries were similar in age, gestation at recruitment and other baseline characteristics but there were differences in ethnicity (In South Africa, 64.0% Black, 35.3%

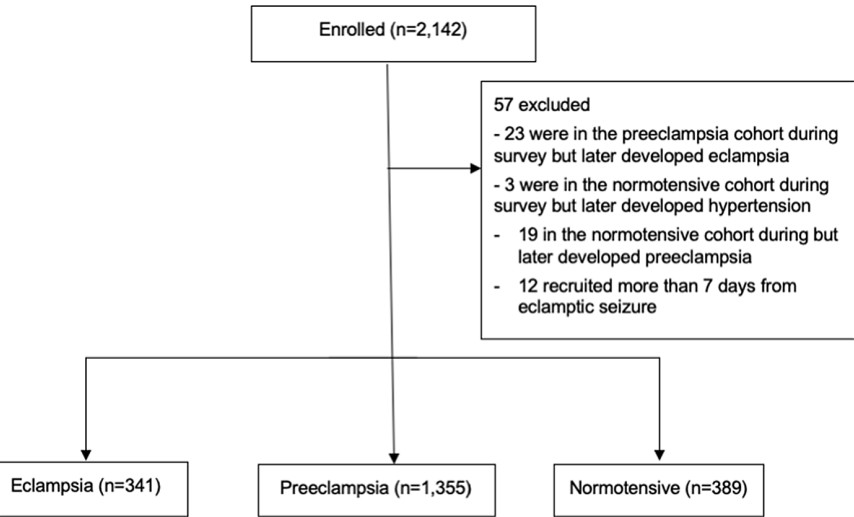

**Fig 1. Recruitment of participants.**

**Table 1. Participant characteristics among pregnancies complicated by preeclampsia, eclampsia, or neither (normotensive pregnancies).** Continuous data are presented as mean and standard deviation (SD) or median and interquartile range (IQR) based on distribution of data. Categorical data are presented as frequency and percentages. *263 participants were recruited postpartum. ^1 case of antepartum maternal death where the foetus remained undelivered.

| Characteristic | Eclampsia (N=341) | Preeclampsia (N=1,355) | Normotensive (N=389) |
|---|---|---|---|
| **Maternal Characteristics** | | | |
| **Maternal age, mean (SD)** | 25.0 (6.1) | 28.4 (6.2) | 28.6 (6.1) |
| Missing | 0 (0%) | 0 (0%) | 2 (0.5%) |
| **Race n (%)** | | | |
| Black | 109 (32.0) | 625 (46.1) | 196 (50.4%) |
| Bi-racial | 48 (14.1) | 316 (23.3) | 149 (38.3%) |
| White | – | 5 (0.4) | 3 (0.8%) |
| Indian | – | 1 (0.07) | 0 (0%) |
| South Asian | 184 (54.0) | 408 (30.1) | 41 (10.5%) |
| **Country n (%)** | | | |
| Pakistan | 183 (53.7) | 408 (30.1%) | 41 (10.5%) |
| South Africa | 158 (46.3) | 947 (69.9%) | 348 (89.5%) |
| **Body mass index, mean (SD)** | 27.0 (5.7) | 29.6 (7.3) | 28.1 (7.5) |
| Missing | 27 (7.9%) | 75 (5.5%) | 24 (6.2%) |
| **Gestation at inclusion (week; median (IQR))*** | 34.8 (31.0, 37.1) | 34.9 (31.7, 37.4) | 34.7 (29.7, 38.6) |
| Missing | 32 (9.4%) | 73 (5.4%) | 10 (2.6%) |
| **Nulliparous n (%)** | 211 (61.9%) | 599 (44.2%) | 112 (28.8%) |
| Missing | 0 (0%) | 1 (0.07%) | 0 (0%) |
| **Antenatal care n (%)** | | | |
| None | 120 (35.2%) | 234 (17.3%) | 16 (4.1%) |
| After 20 weeks | 96 (28.1%) | 358 (26.4%) | 111 (28.5%) |
| Before 20 weeks | 122 (35.8%) | 753 (55.6%) | 258 (66.3%) |
| Missing | 3 (0.9%) | 10 (0.7%) | 4 (1.0%) |
| **Highest systolic blood pressure (median (IQR))** | 170 (160, 185.5) | 165 (153, 180) | 120 (110.0, 126.0) |
| **Highest diastolic blood pressure (median (IQR))** | 110 (100, 120) | 101 (95, 110) | 70 (62, 76) |
| Missing | 1 (0.3%) | 4 (0.3%) | 19 (4.9%) |
| **Anti-hypertensive treatments at booking (%)** | 33 (9.7%) | 140 (10.3%) | 0 (0%) |
| **Tobacco use (%)** | 17 (5.0%) | 88 (6.5%) | 51 (13.1%) |
| **Neonatal Characteristics** | | | |
| **Stillborn (%)** | 57 (16.7%) | 113 (8.3%) | 10 (2.6%) |
| Missing | 2 (0.6) | 92 (6.8%) | 8 (2.1%) |
| **Sex (%)** | | | |
| Female | 162 (47.5) | 650 (48.0) | 209 (53.6%) |
| Missing | 6 (1.8%) | 127 (9.4%) | 8 (2.1%) |
| **Birthweight (median (IQR))** | 2152.5 (1,500, 2,800) | 2080 (1,370, 2,730) | 2,900 (2,170, 3,275) |
| Missing | 11 (3.2%) | 128 (9.4%) | 12 (3.1%) |
| **Gestational age at birth (median (IQR))** | 35.3 (31.9, 37.7) | 35.1 (32.0, 37.6) | 38.1 (34.8, 39.3) |
| Missing | 70 (20.5%) | 134 (9.9%) | 13 (3.3%) |
| **Mode of birth (%)^** | | | |
| Vaginal birth | 87 (25.5%) | 318 (23.5%) | 214 (55.0%) |
| Cesarean section | 249 (73.0%) | 923 (68.1%) | 168 (43.2%) |
| Missing | 4 (1.2%) | 114 (8.4%) | 7 (1.8%) |

Bi-racial, 0.6% White whereas in Pakistan, 100% were South Asian) and the numbers that did not receive antenatal care (6.6% in South Africa versus 43.4% in Pakistan).

Our primary comparison was the likelihood of symptoms being present before an eclamptic seizure versus being present among women with preeclampsia (Table 2). The odds ratio (OR) for a headache preceding an eclamptic seizure was 2.56 (95% confidence interval (CI) [1.81,3.62]; $p<0.001$; 88.0% eclampsia versus 74.1% preeclampsia)—Table 2. There was a stepwise increase in the odds ratios with increasing headache severity (Table D in S1 File). A sudden headache was associated with a higher odds ratio (OR 7.86, 95% CI [5.21,11.88]; $p<0.001$) compared to a headache of gradual onset (OR 1.79, 95% CI [1.25,2.56]; $p=0.0010$).

The odds ratio of visual disturbances preceding eclampsia was 5.73 (95% CI [4.44,7.39]; $p<0.001$,65.1% eclampsia versus 24.6% preeclampsia) and visual disturbances most often occurred within 24 hours of an eclamptic seizure (Table D in S1 File). The odds ratio for epigastric pain was 2.25 (95% CI [1.76,2.89]; $p<0.001$; 41.9% eclampsia versus 24.3% preeclampsia). There was modest increase in the odds ratio with increasing severity of pain (Table D in S1 File). Headache, visual changes, and epigastric pain were uncommon among normotensive pregnancies: headache 3.3%, visual disturbance 0.5% and epigastric pain 1.0% (Table E in S1 File).

We identified 10 prodromal symptoms that were present preceding eclampsia with odds ratios over 10, compared with being present with preeclampsia (Table 2). OR over 30 were: 42.03 for twitching or jerking (95% CI [23.66,74.68]; $p<0.001$; 30.5% present prior to eclampsia versus 1.0% present with preeclampsia); 36.00 for affected hearing (95% CI [18.34,70.65]; $p<0.001$; 21.1% eclampsia versus 0.7% preeclampsia), 33.60 for affected mind state (95% CI [21.39,52.78]; $p<0.001$; 38.7% eclampsia versus 1.8% preeclampsia) and 33.12 for affected speech (95% CI [19.46,54.37]; $p<0.001$; 29.6% eclampsia versus 1.2% preeclampsia). Odds ratios over 20 were: 26.59 for severe vertigo (95% CI [7.82,90.41]; $p<0.001$; 5.6% eclampsia versus 0.2% preeclampsia), 23.71 for feelings of doom (95% CI [16.49,34.10]; $p<0.001$; 45.4% eclampsia versus 3.4% preeclampsia), and 20.52 for confusion (95% CI [14.22,29.63]; $p<0.001$; 41.3% eclampsia versus 3.3% preeclampsia). Odds ratios between 10 and 20 were for: 18.16 (95% CI [10.76,30.66]; $p<0.001$) for jitters/nervous shaking; 15.18 (95% CI [11.34,20.33]; $p<0.001$) for difficulty concentrating; and 10.49 (95% CI [6.76,16.27]; $p<0.001$) for weakness or paralysis.

**Table 2. Odds ratios for the presence of symptoms being present prior to the onset of eclampsia compared to being present among women with preeclampsia. The likelihood of symptoms occurring before eclampsia, compared to being present with preeclampsia was estimated via univariate logistic regression.**

| Symptom | Eclampsia (N=341) | Preeclampsia (N=1,355) | Odds Ratio (95% confidence interval) |
|---|---|---|---|
| **Known prodromal symptoms – N (%)** | | | |
| Headache | 300 (88.0) | 1,004 (74.1) | 2.56 [1.81,3.62] |
| Visual disturbances | 222 (65.1) | 333 (24.6) | 5.73 [4.44,7.39] |
| Epigastric pain | 143 (41.9) | 329 (24.3) | 2.25 [1.76,2.89] |
| **New prodromal symptoms – N (%)** | | | |
| Twitching or jerking of arms or legs | 104 (30.5) | 14 (1.0) | 42.03 [23.66,74.68] |
| Hearing affected | 72 (21.1) | 10 (0.7) | 36.00 [18.34,70.65] |
| Mind state affected | 132 (38.7) | 25 (1.8) | 33.60 [21.39,52.78] |
| Speech affected | 101 (29.6) | 17 (1.2) | 33.12 [19.46,54.37] |
| Severe vertigo | 19 (5.6) | 3 (0.2) | 26.59 [7.82,90.41] |
| Feelings of impending doom or end of the world | 155 (45.4) | 46 (3.4) | 23.71 [16.49,34.10] |
| Confusion | 141 (41.3) | 45 (3.3) | 20.52 [14.22,29.63] |
| Jitters/nervousness/nervous shaking | 70 (20.5) | 19 (1.4) | 18.16 [10.76,30.66] |
| Difficulty concentrating | 192 (56.3) | 106 (7.8) | 15.18 [11.34,20.33] |
| Weakness or paralysis | 69 (20.2) | 32 (2.4) | 10.49 [6.76,16.27] |

When the data were split according to whether they were collected in Pakistan or South Africa, all 10 symptoms retained odds ratios above 10 at both sites, except weakness or paralysis (OR 12.53, 95% CI [6.60,23.79]) in South Africa and odds ratio 7.07 (95% CI [3.85,13.00]) in Pakistan; Fig 2). Conversely, the known prodromal symptoms of headache, visual changes, and epigastric pain all had odds ratios less than 10 at both sites.

These novel prodromal symptoms rarely occurred among those who had normotensive pregnancies. Among all those with normotensive pregnancies, twitching and jerking, or confusion occurred once, and the remaining top prodromal symptoms were not reported at all (Table E in S1 File). The only odds ratios estimable, for twitching and jerking among those who had eclampsia compared to those with normotensive pregnancies, was 170.26 (95% CI [23.60,1228.21]; $p < 0.001$; 30.5% eclampsia versus 0.3% normotensive).

We identified other novel prodromal symptoms more likely to be present with eclampsia compared with preeclampsia, but with lower odds ratios. They were severe dizziness, feeling anxious, changes in mood, nausea and vomiting, tightness in the chest, oedema or shortness of breath, and focal neurological deficits not including vertigo (Table D in S1 File). They were uncommon among women with normotensive pregnancies (Table E in S1 File).

Eclampsia was uncommon without prodromal symptoms. Only 2.4% (8/341) of those who had eclampsia did not report any of the 20 prodromal symptoms we screened for in this study (Table 3). When confining our analysis to 13 prodromal symptoms (headache, visual changes, epigastric pain, plus the 10 symptoms with odds ratios >10), eclampsia occurred in the absence of these symptoms in only 6.1% of women (21/341; Table 3). Eclampsia was more often associated with the presence of many co-existing prodromal symptoms, compared to preeclampsia (Table 3). Four or more prodromal symptoms were present in 60.5% of those who had eclampsia but only 6.2% who had preeclampsia. Conversely, 2 symptoms or less occurred in 27.8% of those who had eclampsia but 85.1% among those who had preeclampsia.

We did a post-hoc analysis of 23 participants we excluded from the main analysis as they represented a cross over cohort (they were grouped into the preeclampsia cohort at the time of the interview but had an eclamptic seizure subsequently). In this small cohort who developed eclampsia later, four novel symptoms had odds ratios between 5 and 20 (compared to preeclampsia without seizures). The odds ratios for headache and visual changes were not significantly changed (Table F in S1 File).

We did a second post-hoc analysis to examine whether maternal demographics confounded the associations we report between symptoms and eclampsia. OR for novel prodromal symptoms and known symptoms did not materially change when we corrected for maternal age, body mass index and country of recruitment (Table G in S1 File). The odd ratios for

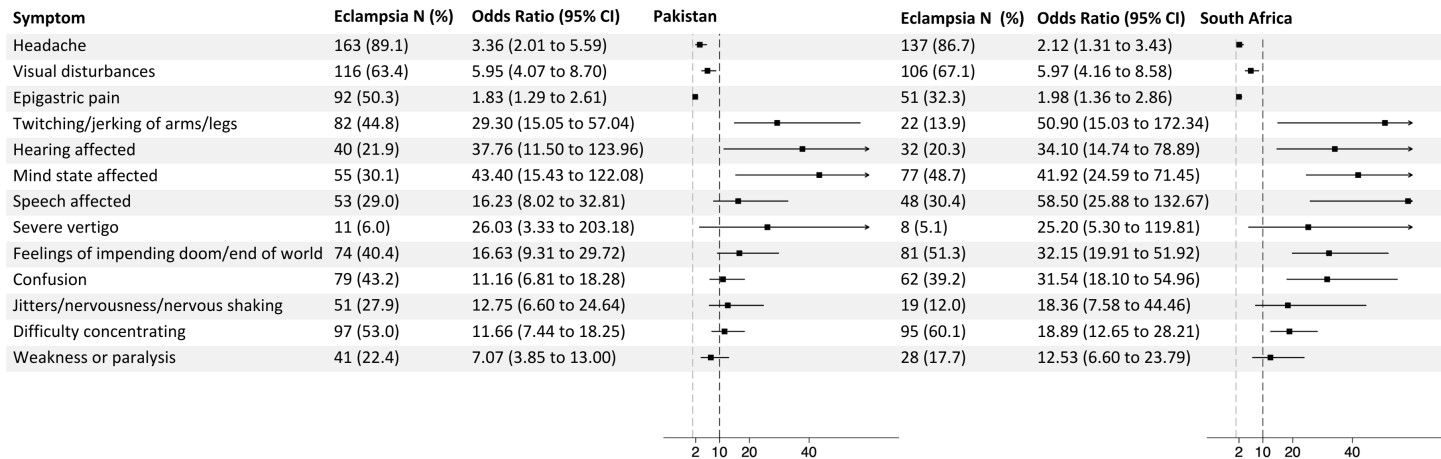

| Symptom | Eclampsia N (%) | Odds Ratio (95% CI) | Pakistan | Eclampsia N (%) | Odds Ratio (95% CI) | South Africa |
|---|---|---|---|---|---|---|
| Headache | 163 (89.1) | 3.36 (2.01 to 5.59) | | 137 (86.7) | 2.12 (1.31 to 3.43) | |
| Visual disturbances | 116 (63.4) | 5.95 (4.07 to 8.70) | | 106 (67.1) | 5.97 (4.16 to 8.58) | |
| Epigastric pain | 92 (50.3) | 1.83 (1.29 to 2.61) | | 51 (32.3) | 1.98 (1.36 to 2.86) | |
| Twitching/jerking of arms/legs | 82 (44.8) | 29.30 (15.05 to 57.04) | | 22 (13.9) | 50.90 (15.03 to 172.34) | |
| Hearing affected | 40 (21.9) | 37.76 (11.50 to 123.96) | | 32 (20.3) | 34.10 (14.74 to 78.89) | |
| Mind state affected | 55 (30.1) | 43.40 (15.43 to 122.08) | | 77 (48.7) | 41.92 (24.59 to 71.45) | |
| Speech affected | 53 (29.0) | 16.23 (8.02 to 32.81) | | 48 (30.4) | 58.50 (25.88 to 132.67) | |
| Severe vertigo | 11 (6.0) | 26.03 (3.33 to 203.18) | | 8 (5.1) | 25.20 (5.30 to 119.81) | |
| Feelings of impending doom/end of world | 74 (40.4) | 16.63 (9.31 to 29.72) | | 81 (51.3) | 32.15 (19.91 to 51.92) | |
| Confusion | 79 (43.2) | 11.16 (6.81 to 18.28) | | 62 (39.2) | 31.54 (18.10 to 54.96) | |
| Jitters/nervousness/nervous shaking | 51 (27.9) | 12.75 (6.60 to 24.64) | | 19 (12.0) | 18.36 (7.58 to 44.46) | |
| Difficulty concentrating | 97 (53.0) | 11.66 (7.44 to 18.25) | | 95 (60.1) | 18.89 (12.65 to 28.21) | |
| Weakness or paralysis | 41 (22.4) | 7.07 (3.85 to 13.00) | | 28 (17.7) | 12.53 (6.60 to 23.79) | |

**Fig 2. Odds ratios for the presence of symptoms prior to the onset of eclampsia compared to being present among women with preeclampsia, divided according to country of recruitment.**

**Table 3. Frequency of 13 symptoms being present in the preeclampsia, eclampsia, and normotensive cohorts. Symptoms were headache, visual changes, epigastric pain, twitching/limb jerking, affected hearing, altered mind state, severe vertigo, impaired speech, feelings of doom, confusion, jitters, difficulty concentrating, and weakness/paralysis. Numbers and percentages shown.**

| Number of symptoms | Eclampsia (*N*=341) | Preeclampsia (*N*=1,355) | Normotensive (*N*=389) |
|---|---|---|---|
| 0 | **21 (6.1%)** | **278 (20.5%)** | **371 (95.4%)** |
| 1 | 23 (6.7%) | 554 (40.9%) | 15 (3.9%) |
| 2 | 51 (15.0%) | 321 (23.7%) | 2 (0.5%) |
| 3 | 40 (11.7%) | 118 (8.7%) | 1 (0.3%) |
| 4 | 34 (10.0%) | 40 (3.0%) | 0 (0%) |
| 5 | 32 (9.4%) | 16 (1.2%) | 0 (0%) |
| 6 | 27 (7.9%) | 14 (1.0%) | 0 (0%) |
| 7 | 21 (6.2%) | 4 (0.3%) | 0 (0%) |
| 8 | 29 (8.5%) | 5 (0.4%) | 0 (0%) |
| 9 | 24 (7.0%) | 3 (0.2%) | 0 (0%) |
| 10 | 18 (5.3%) | 0 (0%) | 0 (0%) |
| 11 | 11 (3.2%) | 2 (0.2%) | 0 (0%) |
| 12 | 4 (1.2) | 0 (0%) | 0 (0%) |
| 13 | 6 (1.8) | 0 (0%) | 0 (0%) |

the novel prodromal symptoms ranged between 9.33 and 40.98 whereas odds ratios for the known symptoms of headache, visual changes, and epigastric pain ranged between 1.8 and 6.3.

## Discussion

It is uncommon to uncover new symptoms for any condition, especially for afflictions that have been known about for centuries [14]. In this prospective study done in two regions of the globe, we identified 10 new prodromal symptoms strongly associated with eclampsia, with odds ratios over 10 when compared to women with preeclampsia. These new symptoms were more strongly associated with eclampsia than headache, visual changes or epigastric pain, prodromal symptoms widely used in the clinic as part of daily history taking from women with preeclampsia, or hypertension during pregnancy. The presence of an increasing number of symptoms made eclampsia more likely. Conversely, the occurrence of eclampsia without prodromal symptoms was infrequent.

Incorporating these 10 symptoms as part of history taking may be useful as part of clinical management of women with preeclampsia or hypertension during pregnancy. Their presence may suggest preeclampsia with neurological involvement. Importantly, their presence may mean there is an elevated risk of eclampsia.

While this was not intended to be a diagnostic prediction study, the symptoms we have identified may be useful to direct who to give magnesium sulphate, in the same way existing symptoms (with far lower odds ratios) are currently used [15–17]. For instance, the sudden onset of twitching or arm jerking (OR 42.03), affected hearing (OR 36.0), and confusion (OR 20.52) in a woman with preeclampsia might raise concerns that she is at elevated risk of eclampsia, where the commencement of magnesium sulphate is reasonable. In contrast, those with preeclampsia but who are asymptomatic appear to be at very low risk.

Currently, there is no consensus which pregnancies complicated by preeclampsia should be offered magnesium sulphate. Without reliable clinical indicators to predict eclampsia, recommendations from international guidelines vary widely regarding who to offer magnesium sulphate. Some recommend magnesium sulphate if prodromal symptoms (specifically headache, visual changes, or epigastric pain) or other features are present, such as severe hypertension [15–17] or if there is progressive deterioration in laboratory blood tests [16,17]. Others do not base their recommendations on the presence of symptoms: The American College of Obstetricians and Gynecologists (ACOG) guidelines recommend magnesium

sulphate is offered to women with preeclampsia and severe features [18] (this would include headache or visual changes, as these are considered severe features) while The World Health Organisation recommends magnesium sulphate is given to all women with preeclampsia, even mild disease [19].

Classic prodromal symptoms of headache, visual disturbances, and epigastric pain were only modestly associated with eclampsia. Their generally high prevalence in preeclamptic pregnancies may be why they performed modestly. In contrast, the absolute incidence of the 10 new symptoms we report was low with preeclampsia but high with eclampsia.

The absence of any prodromal symptoms may mean the eclampsia risk is very low. Hence, for those with preeclampsia without neurological symptoms, it may be reasonable to withhold magnesium sulphate, especially in high income settings where the background prevalence is low [1,6]. Such a selective approach to offering magnesium sulphate based on the presence of symptoms may reduce numbers given the drug and this could reduce healthcare costs.

Our study had several limitations. While we recruited participants prospectively, we asked about prodromal symptoms after the seizure happened. However, a prospective screening study to find new symptoms would have been highly challenging. Eclampsia is rare, even in low-middle income settings meaning extremely large numbers would need to be screened. Another limitation is that clinicians may have commenced magnesium sulphate if the classic prodromal symptoms of headache and visual disturbances were present, but not for the novel symptoms we have identified. This may have introduced interventional bias and reduced the apparent strength of association with eclampsia for these known prodromal symptoms.

Another potential limitation is the risk of recall bias. Those affected by eclampsia will have had a significant neurological episode meaning they might have recalled symptoms differently from those with preeclampsia or were normotensive. However, the relative differences in odds ratios between symptoms within groups may still be valid for two reasons. First, the entire symptom list was screened within groups meaning all symptoms would be equivalently prone to recall bias. Second, the novel prodromal symptoms had consistently higher odds ratios compared with known symptoms when the data were split according to country collected (Fig 2), and this provides validity.

Additionally, recall periods may have differed between groups, with most women with eclampsia recruited on the day of their seizure and asked to recall symptoms experienced before seizure onset, whereas women in the control groups were asked to recall symptoms from the preceding 7 days before consent. This difference may introduce differential misclassification.

Ideally, the associations between symptoms and eclampsia are validated in other populations and in different settings. Given we have recruited women from three tertiary centres it is critical that applicability across care settings is assessed, specifically in peripheral and community health services. Future studies must also consider how and when these symptoms should be assessed and implemented into antenatal care.

Strengths of our study were that we recruited participants within seven days of an eclamptic seizure or while they still had preeclampsia, that a research midwife interviewed the participants, and we used the same questionnaire for both countries. We were agnostic as to which novel symptom could be associated with eclampsia, which minimised bias.

The new symptoms we identified do not localise to a specific region of the brain such as the posterior compartment, suggesting the pathophysiology of eclampsia is diffuse. This is in keeping with eclampsia being a maternal vascular lesion that impairs vessel autoregulation and can affect any region of the brain [1,20,21]. Also, the prevalence of the novel symptoms we described was high in the eclamptic cohort, less in the preeclampsia cohort but notably, hardly occurred in normotensive pregnancies. This hints at a biological gradient where some degree of neuronal pathology exists in cases of preeclampsia when there are neurological symptoms, even in the absence of seizures.

In summary, in this two-country prospective study, we identified new prodromal symptoms strongly associated with eclampsia. Eclampsia was uncommon in the absence of any prodromal symptoms. These new symptoms we have uncovered may be useful as part of history taking to more accurately assess likely eclampsia risk. It could be considered as part of the whole clinical picture to judge whether delivery is indicated in cases of preterm preeclampsia. Finally, these symptoms may be useful in deciding whether magnesium sulphate is required or could be safely withheld.

## Supporting information

**S1 File. Supplementary tables. Table A:** Symptoms asked in the survey. **Table B:** Additional participant characteristics (not listed in Table 1 of the main paper) in the pregnancies affected by preeclampsia, eclampsia, or neither (normotensive pregnancies). **Table C:** Participant characteristics and pregnancy outcomes by study site. **Table D:** The incidence of various symptoms and the odds of subsequently developing eclampsia – Full list of questions on symptoms screened. **Table E:** The incidence of various symptoms in the normotensive and eclampsia cohorts, and the odds of subsequently developing eclampsia. **Table F:** Likelihood of developing eclampsia comparing women recruited with preeclampsia and subsequently developed eclampsia to those with preeclampsia only. **Table G:** Adjusted odds ratio for the likelihood of developing eclampsia compared to those with preeclampsia.
(DOCX)

**S1 Protocol. Study protocol.**
(DOCX)

**S2 File. Primary data (available for download).**
(XLS)

**S1 STROBE Checklist. The STROBE checklist is best used in conjunction with this article (freely available on the websites of PLoS Medicine at http://www.plosmedicine.org/, Annals of Internal Medicine at http://www.annals.org/, and Epidemiology at http://www.epidem.com/).** Information on the STROBE Initiative is available at www.strobe-statement.org. DOI: https://doi.org/10.1136/bmj.39335.541782.AD.
(DOCX)

## Acknowledgments

We thank all the women who participated in this study, the staff at Fatima Memorial Hospital and Jinnah Hospital in Lahore, Pakistan; and Tygerberg Hospital in Cape Town, South Africa.

## Author contributions

**Conceptualization:** Roxanne Hastie, Stephen Tong.

**Data curation:** Roxanne Hastie, Farhatulain Ahmed, Parinaz Mehdipour, Jacqui Visser, Anam Bashir, Jessica A. Atkinson, Catherine Cluver, Lina Bergman.

**Formal analysis:** Roxanne Hastie, Parinaz Mehdipour, Lyle Gurrin, Jessica A. Atkinson.

**Funding acquisition:** Roxanne Hastie.

**Investigation:** Roxanne Hastie, Jacqui Visser, Anam Bashir, Anthea Lindquist, Catherine Cluver, Lina Bergman, Stephen Tong, Farhatulain Ahmed.

**Methodology:** Roxanne Hastie, Bernard Yan, Lyle Gurrin, Anthea Lindquist, Catherine Cluver, Lina Bergman, Susan P. Walker.

**Project administration:** Roxanne Hastie, Farhatulain Ahmed, Susan P. Walker, Jacqui Visser, Stephen Tong, Catherine Cluver, Lina Bergman.

**Resources:** Roxanne Hastie, Stephen Tong, Anthea Lindquist.

**Supervision:** Roxanne Hastie, Farhatulain Ahmed, Susan P. Walker, Catherine Cluver, Lina Bergman, Stephen Tong.

**Validation:** Roxanne Hastie.

**Writing – original draft:** Roxanne Hastie, Stephen Tong.

**Writing – review & editing:** Roxanne Hastie, Farhatulain Ahmed, Parinaz Mehdipour, Bernard Yan, Susan P. Walker, Jacqui Visser, Anam Bashir, Lyle Gurrin, Anthea Lindquist, Jessica A. Atkinson, Catherine Cluver, Lina Bergman, Stephen Tong.

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
