## [Editor Report · Decision Letter 0]

1 Sep 2025

Dear Dr Tong,

Thank you for submitting your manuscript entitled “Identifying novel prodromal symptoms of eclampsia: A multi-national, case-cohort study of prospectively collected data” for consideration by PLOS Medicine.

Your manuscript has now been evaluated by the PLOS Medicine editorial staff and I am writing to let you know that we would like to send your submission out for external assessment.

However, we first need you to complete your submission by providing the metadata that are required for full assessment. To this end, please login to Editorial Manager where you will find the paper in the ‘Submissions Needing Revisions’ folder on your homepage. Please click ‘Revise Submission’ from the Action Links and complete all additional questions in the submission questionnaire.

For clinical studies, please upload a copy of your trial study protocol as a supporting information file. The study protocol should be the version submitted for approval to the institutional review board or ethics committee, should include any amendments to the study protocol, as well as the date of their approval by the institutional review or ethics committee. Please also detail any deviations from the study protocol in the Methods section of your manuscript. The editors will consider the protocol and study conduct prior to a final decision for external review.

Please re-submit your manuscript within two working days, i.e. by Sep 03 2025 11:59PM.

Once your full submission is complete, your paper will undergo a series of checks in preparation for full assessment.

Kind regards,

Richard Turner, PhD

Consulting Editor, PLOS Medicine

plosmedicine@plos.org

---

## [Decision Letter · Decision Letter 1]

5 Nov 2025

Dear Dr Tong,

Many thanks for submitting your manuscript “Identifying novel prodromal symptoms of eclampsia: A multi-national, case-cohort study of prospectively collected data” (PMEDICINE-D-25-03006R1) to PLOS Medicine. The paper has been reviewed by subject experts and a statistician; their comments are included below and can also be accessed here: [LINK]

As you will see, the reviewers thought your study is interesting, but also have some concerns regarding the methodology. They had concerns about the lack of adjustment for confounders, would like an explanation why the specific symptoms were chosen, and had concerns about the possibility of recall bias, and the lack of antenatal care. After discussing the paper with the editorial team and an academic editor with relevant expertise, I’m pleased to invite you to revise the paper in response to the reviewers’ comments. We plan to send the revised paper to some or all of the original reviewers, and we cannot provide any guarantees at this stage regarding publication.

We ask that you submit your revision by Nov 26 2025 11:59PM. However, if this deadline is not feasible, please contact me by email, and we can discuss a suitable alternative.

Don’t hesitate to contact me directly with any questions (sbruijn@plos.org).

Best regards,

Suzanne

Suzanne De Bruijn, PhD

Associate Editor

PLOS Medicine

sbruijn@plos.org

Comments from the reviewers:

Reviewer #1: This is a case-control study that aimed to identify risk factors of eclampsia compared with preeclampsia and normotensive pregnancies. I have several comments on the methods and statistics:

1. This is a straightforward case-control study and the abstract and main text reflect so. However the title stated ‘case-cohort study’ which is not really a commonly used name for epidemiological design. Can the authors please clarify.

2. Preeclampsia is defined as hypertension plus at least 1 symptom while normotensive is defined as without hypertension or eclampsia. What about pregnancies that had hypertension but no other symptoms? Were they excluded?

3. The symptom recall period of the 3 groups were different - for eclampsia pregnancy the recall seems to be longer as that have to be occurred prior to seizure onset. Could the authors clarify what is the median and IQR duration between seizure and enrolment? Either way the potential of differential misclassification should be discussed in detail.

4. Comparison between groups were made in univariate logistic regression models and no adjustment of confounders were made. Can the authors justify this approach please? This is really unusual because it is subjected to confounding (e.g. age, smoking, and SES could be a cause for both the symptoms (or the recall of it) and outcome).

5. The discussions seem to around the use of these symptoms to identify women at higher risk of eclampsia - which essentially is risk prediction. However, the authors’ analyses could not support the usefulness of these symptoms for risk predictions. Something like AUC based on a contemporary variable selection approach (e.g. LASSO) would be required to justify their use in clinical practice.

Generally this study appears to have some value but the analysis at the moment appears to be too simplistic to be useful. In addition, the difference in recall between groups could lead to severe recall bias.

Reviewer #2: Hastie et al. describe in their study a prospective case-control study in South Africa and Pakistan to identify new prodromal symptoms of preeclampsia. Their findings suggest only few patients that will experience eclampsia will not have any prodromal symptoms and describe 10 neurological symptoms which appear prior to the eclampsia in the majority of the patients. They suggest screening for these prodromal symptoms in patients with preeclampsia to guide management regarding magnesium sulphate prophylaxis.

An aspect which remains somewhat worrying is that participants were asked about prodromal symptoms after the seizure happened. All participants were questioned at study entry, but it is likely that only in case of a seizure women may actively remember such symptoms. In the absence of a serious event one is bound to also not come up with all kinds of potential symptoms. This causes severe bias.

Could the authors elaborate on their rationale to use these specific prodromal symptoms?

The authors state in the methods that in the South African cohort proteinuria had to be present for the diagnosis of preeclampsia? Does this mean that in South Africa preeclampsia was not defined according to the ISSHP 2018 guidelines?

Why were women with preeclampsia who developed eclampsia and undertook the survey not included? As the goal of this study was to identify prodromal symptoms for development of eclampsia in patients with preeclampsia, this would be the patients of interest? Especially because the way the data is presented, recall bias is an important limitation and the patients that have now been excluded in the analysis (N=23, which is a relatively large amount of patients given the rarity of eclampsia) would give insight in prodromal symptoms without recall bias.

Why do the authors declare that there were 12 patients that filled out the questionnaire more than 7 days ago? In the Methods section it is stated that patients patients that developed eclampsia in the last 7 days were enrolled, so is filling out the questionnaire after more than 7 days an exclusion criterium?

Given the setting of these cohorts, how generalizable are these findings to other settings? Almost half of all patients that were included in Pakistan did not receive antenatal care, which is likely to play an important role in the timely identification of symptoms and timely adequate management of e.g. preceding pregnancy induced hypertension, or even timely induction and delivery before eclampsia develops. How do the authors feel about the importance of addressing the cause of the increased incidence of eclampsia in these areas rather than screening for prodromal symptoms?

Can the authors provide any details on how the included women presented themselves in the hospital? How many patients experienced eclampsia outside of the hospital and is there a window of opportunity for screening with these prodromal symptoms in the researched settings?

Have the authors only asked patients for these specific symptoms or have they also asked what symptoms the patients experienced in an open question? And if so: were there any insights here?

The Magpie Trial (Duley et al., 2002, Lancet, DOI: 10.1016/s0140-6736(02)08778-0) provides evidence that magnesium sulphate halves the risk of eclampsia in patients with preeclampsia (pregnant or max. 24h postpartum) with a diastolic blood pressure and uncertainty about whether to use magnesium sulphate. This is the only evidence regarding treatment for preeclampsia. Can the authors elaborate how the screening for prodromal symptoms adds to this research? Do the authors feel that the prodromal symptoms should be considered rather than administering magnesium sulphate to all patients with uncertainty about the use of magnesium sulphate to prevent adverse outcomes, given limited side effects? Indeed, what would be the next step: screen prospectively on the basis of these symptoms, or perhaps define a scoring system where a certain minimum number of symptoms needs to be present? Might this be designed based on the current data? This will help to determine whether these symptoms are really eclampsia-specific. In other words, what are the practical consequences. If none of this is planned, the significance of these findings diminishes. Please also consider costs.

Reviewer #3: Overall:

This is a very interesting and impressive study which has the potential to alter clinical practice and guidelines. As the authors state, it is uncommon to uncover new symptoms for any condition, and this manuscript highlights the importance of continuing to question current practice and improve the body of established medical evidence - this is even more important in the often-neglected field of women’s health and this study contributes important, novel, findings to our understanding of a condition which remains a leading cause of maternal death globally. As an Obstetrician, I believe the findings offer real clinical utility; it would be great to understand more about how they could be taken further and incorporated into routine practice, as well as evaluated more widely and across different populations and settings. The identification of these 10 prodromal symptoms which may predict the onset of eclampsia has the potential to offer a simple, low-cost screening tool that could prompt earlier intervention and thus reduce maternal and perinatal mortality. There are a few points below which I feel could benefit from further detail, as well as two small typos which I have highlighted on the pdf.

Abstract:

1. Clear and informative

Introduction:

1. Also very clear and informative, clearly sets out the research question and provides helpful context, with concise definitions.

Methods:

1. It would be interesting to know why the two sites were selected, and if word count allowed, a brief description of the maternity units (e.g. number of deliveries, population served, world-bank income classification and health system structure e.g. publicly or privately funded, insurance-based etc.).

2. I’m also interested to know why a period of 7 days was chosen, though this seems logical given the clinical nature of pre-eclampsia and eclampsia and the need to balance contemporaneous reviews in a high-risk postnatal population whilst also allowing time for recovery

3. Was a sample size calculation performed?

4. I understand that the symptoms were discussed with a neurologist, however it would be interesting (especially for the non-neurologist) to understand why these particular 20 symptoms were chosen, and any evidence/literature justifying them.

5. Were there any issues in translating the interview questions, and can the team be sure that the symptoms were well-understood by participants particularly if speaking a different language and accounting for a different cultural context

6. How were the diagnoses of pre-eclampsia and eclampsia verified by the clinical teams?

Results:

1. 35.2% of women in the eclamptic cohort received no antenatal care; this lack of surveillance and potential earlier detection of pre-eclampsia (and eclampsia) may have contributed to the severity of their disease and presence of severe symptoms: it would be interesting to repeat this study in an even larger cohort of women across multiple different care settings, in order to establish whether screening for these prodromal symptoms is useful in all settings, or only those were there is a limited antenatal care and therefore later presentations to care with more severe disease.

2. It would also be interesting to establish whether symptom severity correlates with blood pressure and other markers of pre-eclampsia severity (biochemistry, ultrasound findings)

3. To have evaluated a cohort of 341 women who experienced eclampsia is hugely impressive - I hope that the editors appreciate that this is a relatively rare outcome, and thus to have a cohort of 341 women is excellent

Discussion:

1. The authors state that the ten symptoms identified as being strongly associated with eclampsia may be useful as part of clinical management of women with pre-eclampsia. I think it would be interesting to better understand how the authors envisage this could be incorporated into routine clinical practice. Do they propose a routine checklist at antenatal visits? Do they plan on assessing the feasibility of this? Have they undertaken any qualitative assessment of healthcare providers views on such a checklist and whether they feel it would be a helpful part of their assessment?

2. I also think it would be useful to evaluate these symptoms across a broader range of settings in order to establish the universality of such symptoms, and whether screening is more or less useful according to different settings with differing resources available.

3. A huge strength of this study, in my opinion, is the potential for a very simple, low-cost screening tool which could be implemented in low and middle income countries and potentially prompt earlier intervention. It would be useful to have more information about whether the authors feel there is sufficient evidence from their findings to support this, and whether there is scope to further evaluate this as a screening tool across other settings. I would be interested to explore this in Zambia and Sierra Leone.

Alice Beardmore-Gray

---

* Please upload any figures associated with your paper as individual TIF or EPS files with 300dpi resolution at resubmission; please read our figure guidelines for more information on our requirements: http://journals.plos.org/plosmedicine/s/figures. While revising your submission, we strongly recommend that you use PLOS’s NAAS tool (https://ngplosjournals.pagemajik.ai/artanalysis) to test your figure files. NAAS can convert your figure files to the TIFF file type and meet basic requirements (such as print size, resolution), or provide you with a report on issues that do not meet our requirements and that NAAS cannot fix.

After uploading your figures to PLOS’s NAAS tool - https://ngplosjournals.pagemajik.ai/artanalysis, NAAS will process the files provided and display the results in the “Uploaded Files” section of the page as the processing is complete.

If the uploaded figures meet our requirements (or NAAS is able to fix the files to meet our requirements), the figure will be marked as “fixed” above. If NAAS is unable to fix the files, a red “failed” label will appear above.

When NAAS has confirmed that the figure files meet our requirements, please download the file via the download option, and include these NAAS processed figure files when submitting your revised manuscript.

* Please ensure that the paper adheres to the PLOS Data Availability Policy (see http://journals.plos.org/plosmedicine/s/data-availability), which requires that all data underlying the study’s findings be provided in a repository or as Supporting Information. For data residing with a third party, authors are required to provide instructions with contact information (web or email address) for obtaining the data. Please note that a study author cannot be the contact person for the data. PLOS journals do not allow statements supported by “data not shown” or “unpublished results.” For such statements, authors must provide supporting data or cite public sources that include it.

* We expect all researchers with submissions to PLOS in which author-generated code underpins the findings in the manuscript to make all author-generated code available without restrictions upon publication of the work. In cases where code is central to the manuscript, we may require the code to be made available as a condition of publication. Authors are responsible for ensuring that the code is reusable and well documented. Please make any custom code available, either as part of your data deposition or as a supplementary file. Please add a sentence to your data availability statement regarding any code used in the study, e.g. “The code used in the analysis is available from Github [URL] and archived in Zenodo [DOI link]” Please review our guidelines at https://journals.plos.org/plosmedicine/s/materials-software-and-code-sharing and ensure that your code is shared in a way that follows best practice and facilitates reproducibility and reuse. Because Github depositions can be readily changed or deleted, we encourage you to make a permanent DOI’d copy (e.g. in Zenodo) and provide the URL.

* Financial disclosure: Please add a URL to the funders website.

* COI: please include that Roxanne Hastie is a statistical reviewer for PLOS Medicine

* Data availability: if data cannot be publicly shared, please explain why not, and how researchers could ask for access to the data. Please be aware that the contact person cannot be a study author.

* Please include an ethics statement in your manuscript. This should include the name of the institutional review board, the IRB approval number, and whether women gave written or oral consent.

* At this stage, we ask that you include a short, non-technical Author Summary of your research to make findings accessible to a wide audience that includes both scientists and non-scientists. The Author Summary should immediately follow the Abstract in your revised manuscript. This text is subject to editorial change and should be distinct from the scientific abstract. Ideally each sub-heading should contain 2-3 single sentence, concise bullet points containing the most salient points from your study. In the final bullet point of ‘What Do These Findings Mean?’, please include the main limitations of the study in non-technical language. Please see our author guidelines for more information: https://journals.plos.org/plosmedicine/s/revising-your-manuscript#loc-author-summary.

* Please express the main results with 95% CIs as well as p values. When reporting p values please report as p<0.001 and where higher as the exact p value p=0.002, for example. Throughout, suggest reporting statistical information as follows to improve clarity for the reader “22% (95% CI [13%,28%]; p</=)”. Please be sure to define all numerical values at first use.

SUPPLEMENTARY MATERIAL

REFERENCES

OBSERVATIONAL STUDIES

* Abstract: Please include the study design, population and setting, number of participants, years during which the study took place (enrollment and follow up), length of follow up, and main outcome measures.

* Please ensure that the study is reported according to the STROBE (or appropriate STOBE extension) guideline (available from: https://www.equator-network.org/reporting-guidelines/strobe) and include the completed STROBE (or STROBE extension) checklist as Supporting Information. Please add the following statement, or similar, to the Methods: “This study is reported as per the Strengthening the Reporting of Observational Studies in Epidemiology (STROBE) guideline (S1 Checklist).” When completing the checklist, please use section and paragraph numbers, rather than page numbers.

* For all observational studies, in the manuscript text, please indicate: (1) the specific hypotheses you intended to test, (2) the analytical methods by which you planned to test them, (3) the analyses you actually performed, and (4) when reported analyses differ from those that were planned, transparent explanations for differences that affect the reliability of the study’s results. If a reported analysis was performed based on an interesting but unanticipated pattern in the data, please be clear that the analysis was data driven.

* Please state in the Methods section whether the study had a prospective protocol or analysis plan. If a prospective analysis plan (from your funding proposal, IRB or other ethics committee submission, study protocol, or other planning document written before analyzing the data) was used in designing the study, please include the relevant document(s) with your revised manuscript as a Supporting Information file to be published alongside your study and cite it in the Methods section. A legend for this file should be included at the end of your manuscript. If no such document exists, please make sure that the Methods section transparently describes when analyses were planned, and when/why any data-driven changes to analyses took place. Changes in the analysis, including those made in response to peer review comments, should be identified as such in the Methods section of the paper, with rationale.

---

## [Decision Letter · Decision Letter 2]

21 Jan 2026

Dear Dr Tong,

Many thanks for submitting your manuscript “Identifying novel prodromal symptoms of eclampsia: A multi-national, case-control study of prospectively collected data” (PMEDICINE-D-25-03006R2) to PLOS Medicine. The paper has been re-reviewed by subject experts and a statistician; their comments are included below and can also be accessed here: [LINK]

Two of the reviewers think this can be accepted, whereas the statistical reviewer asked for an adjusted analysis.

As we discussed per email, I’m returning the manuscript to you to address these remaining comments. We’re happy if you add this as a post hoc analysis and add it to the supplementary materials. We plan to send the revised paper to some or all of the original reviewers, and we cannot provide any guarantees at this stage regarding publication.

We ask that you submit your revision by Feb 11 2026 11:59PM. However, if this deadline is not feasible, please contact me by email, and we can discuss a suitable alternative.

Don’t hesitate to contact me directly with any questions (sbruijn@plos.org).

Best regards,

Suzanne

Suzanne De Bruijn, PhD

Associate Editor

PLOS Medicine

sbruijn@plos.org

---

## [Decision Letter · Decision Letter 3]

13 Feb 2026

Dear Dr. Tong,

Thank you very much for re-submitting your manuscript “Identifying novel prodromal symptoms of eclampsia: A multi-national, case-control study.” (PMEDICINE-D-25-03006R3) for review by PLOS Medicine.

I have discussed the paper with my colleagues and the academic editor and it was also seen again by 1 reviewer. I am pleased to say that provided the remaining editorial and production issues are dealt with we are planning to accept the paper for publication in the journal.

[LINK]

In revising the manuscript for further consideration here, please ensure you address the specific points made by each reviewer and the editors. In your rebuttal letter you should indicate your response to the reviewers’ and editors’ comments and the changes you have made in the manuscript. Please submit a clean version of the paper as the main article file. A version with changes marked must also be uploaded as a marked up manuscript file.

Please also check the guidelines for revised papers at http://journals.plos.org/plosmedicine/s/revising-your-manuscript for any that apply to your paper. If you haven’t already, we ask that you provide a short, non-technical Author Summary of your research to make findings accessible to a wide audience that includes both scientists and non-scientists. The Author Summary should immediately follow the Abstract in your revised manuscript. This text is subject to editorial change and should be distinct from the scientific abstract.

Please note, when your manuscript is accepted, an uncorrected proof of your manuscript will be published online ahead of the final version, unless you’ve already opted out via the online submission form. If, for any reason, you do not want an earlier version of your manuscript published online or are unsure if you have already indicated as such, please let the journal staff know immediately at plosmedicine@plos.org.

We look forward to receiving the revised manuscript by Feb 20 2026 11:59PM.

Sincerely,

Suzanne De Bruijn, PhD

Associate Editor

PLOS Medicine

plosmedicine.org

Requests from Editors:

* Please change your title to: “identifying novel prodromal symptoms of eclampsia: a two-country, case-control study ”.

* Please ensure that all abbreviations are defined at first use throughout the text.

* Please confirm that all numbers presented in the abstract are present and identical to numbers presented in the main manuscript text.

"* Statistical reporting: Please revise throughout the manuscript, including tables and figures.

- Please report statistical information as follows to improve clarity for the reader ““22% (95% CI [13,28]; p</=)””.

- Please separate upper and lower bounds with commas instead of hyphens as the latter can be confused with reporting of negative values.

- Please repeat statistical definitions (HR, CI etc.) for each set of parentheses."

* In the last sentence of the Abstract Methods and Findings section, please describe the main limitation(s) of the study’s methodology.

* Please ensure that the study is reported according to the STROBE guideline, and include the completed STROBE checklist as Supporting Information. Please add the following statement, or similar, to the Methods: ““This study is reported as per the Strengthening the Reporting of Observational Studies in Epidemiology (STROBE) guideline (S1 Checklist).” The STROBE guideline can be found here: http://www.equator-network.org/reporting-guidelines/strobe/ When completing the checklist, please use section and paragraph numbers, rather than page numbers.”

FUNDING STATEMENT

* The funding statement should include: specific grant numbers, initials of authors who received each award, URLs to sponsors’ websites. Also, please state whether any sponsors or funders (other than the named authors) played any role in study design, data collection and analysis, the decision to publish, or preparation of the manuscript. If they had no role in the research, include this sentence: “The funders had no role in study design, data collection and analysis, decision to publish, or preparation of the manuscript.”

AUTHOR SUMMARY

*Why was this study done: point 2: Please revise to “Giving magnesium sulphate can half the risk of eclampsia, but predicting who is at risk of an eclamptic seizure is challenging.”

* “what did the researchers do and find” : there is a typo in ‘Researcher’, please amend.

* “what did the researchers do and find”: please combine point 1 and point 2. We suggest: “We performed a study in Cape Town and Pakistan, in which we asked women who had a recent eclamptic seizure which symptoms they experienced (before the seizure occurred), among a list of novel neurological symptoms, which have not been reported to be associated with eclampsia.’

*What did the researchers do and find, 5th point: We suggest to remove this, as we limit the number of bullet points to 3.

Comments from Reviewers:

Reviewer #1: Thank you for revising the title and conducting the additional analysis - the consistency adds confidence to the findings.

[LINK]

---

## [Editor Report · Decision Letter 4]

3 Mar 2026

Dear Dr Tong,

On behalf of my colleagues and the Academic Editor, Andrew Shennan, I am pleased to inform you that we have agreed to publish your manuscript “Identifying novel prodromal symptoms of eclampsia: A two-country, case-control study.” (PMEDICINE-D-25-03006R4) in PLOS Medicine.

We do have 2 remaining requests:

1) Thank you for including the STROBE checklist. Can you please modify this to include sections and paragraph numbers, rather than page numbers?

2) Please ensure your statistical reporting is done as per PLOS guidelines. I appreciate that this has been modified, but it is not consistent throughout the manuscript. Please change to follow our guidelines where necessary. This includes but is not limited to, the introduction, Table 2, figure 2 (left).

As a reminder:

* Statistical reporting: Please revise throughout the manuscript, including tables and figures.

- Please report statistical information as follows to improve clarity for the reader ““22% (95% CI [13,28]; p</=)””.

- Please separate upper and lower bounds with commas instead of hyphens as the latter can be confused with reporting of negative values.

- Please repeat statistical definitions (HR, CI etc.) for each set of parentheses.

PRESS

Sincerely,

Suzanne De Bruijn, PhD

Associate Editor

PLOS Medicine